# Revised Calculation of Kalinowski's Ancestral and New Inbreeding Coefficients

**Harmen P. Doekes** [1,2,*] **, Ino Curik** [3] **, István Nagy** [4] **, János Farkas** [5] **, György Kövér** [5] **and Jack J. Windig** [1,2]

1   Animal Breeding and Genomics, Wageningen University & Research, P.O. Box 338,
    6700 AH Wageningen, The Netherlands; jack.windig@wur.nl
2   Centre for Genetic Resources the Netherlands, Wageningen University & Research, P.O. Box 16,
    6700 AA Wageningen, The Netherlands
3   Department of Animal Science, Faculty of Agriculture, University of Zagreb, Svetošimunska cesta 25,
    10000 Zagreb, Croatia; icurik@agr.hr
4   Faculty of Agricultural and Environmental Sciences, Kaposvár University, P.O. Box 16,
    H-7400 Kaposvár, Hungary; nagy.istvan@ke.hu
5   Faculty of Economic Science, Kaposvár University, P.O. Box 16, H-7400 Kaposvár, Hungary;
    farkas.janos.51@gmail.com (J.F.); kover.gyorgy@ke.hu (G.K.)
*   Correspondence: harmen.doekes@wur.nl

**Abstract:** To test for the presence of purging in populations, the classical pedigree-based inbreeding coefficient ($F$) can be decomposed into Kalinowski's ancestral ($F_{ANC}$) and new ($F_{NEW}$) inbreeding coefficients. The $F_{ANC}$ and $F_{NEW}$ can be calculated by a stochastic approach known as gene dropping. However, the only publicly available algorithm for the calculation of $F_{ANC}$ and $F_{NEW}$, implemented in GRain v 2.1 (and also incorporated in the PEDIG software package), has produced biased estimates. The $F_{ANC}$ was systematically underestimated and consequently, $F_{NEW}$ was overestimated. To illustrate this bias, we calculated $F_{ANC}$ and $F_{NEW}$ by hand for simple example pedigrees. We revised the GRain program so that it now provides unbiased estimates. Correlations between the biased and unbiased estimates of $F_{ANC}$ and $F_{NEW}$, obtained for example data sets of Hungarian Pannon White rabbits (22,781 individuals) and Dutch Holstein Friesian cattle (37,061 individuals), were high, i.e., >0.96. Although the magnitude of bias appeared to be small, results from studies based on biased estimates should be interpreted with caution. The revised GRain program (v 2.2) is now available online and can be used to calculate unbiased estimates of $F_{ANC}$ and $F_{NEW}$.

**Keywords:** ancestral inbreeding; new inbreeding; purging; gene dropping; inbreeding depression

## 1. Introduction

Inbreeding is the mating between (close) relatives and is unavoidable in genetically small populations. The degree of inbreeding is typically measured with pedigree-based inbreeding coefficients, as introduced by Wright [1]. Individuals with higher inbreeding coefficients show a lower phenotypic performance on average, a phenomenon known as inbreeding depression [2–4]. Inbreeding depression occurs because part of the genetic load in populations, known as inbreeding load, is only expressed in homozygotes [2]. Inbreeding depression is expected to be largely due to partial dominance, i.e., the existence of (partially) deleterious recessive alleles, although overdominance and epistasis may also play a role [2,3,5].

Inbreeding load in a population is not constant, but rather dynamic over time. New deleterious recessive alleles arise continuously by mutation and these alleles are eroded over time by (natural

and/or artificial) selection and genetic drift [2]. Inbreeding increases the efficiency of selection against deleterious recessive alleles in a process called purging [2,6].

To test for the existence of purging in populations, various pedigree-based methods have been proposed [7–9]. To test for purging in captive wildlife populations, Ballou [7] introduced the ancestral inbreeding coefficient, which is the probability that a random allele in an individual has been previously exposed to inbreeding, i.e., that this allele has been identical-by-descent (IBD) in at least one ancestor. While investigating purging in the captive breeding program of the Speke's gazelle (*Gazella Spekei*), Kalinowski et al. [8] extended Ballou's concept by considering the IBD-status of the individual as well. In Kalinowski's approach the total pedigree-based inbreeding coefficient is decomposed into an ancestral (**$F_{ANC}$**) and a new (**$F_{NEW}$**) inbreeding coefficient. The $F_{ANC}$ is the probability that alleles are IBD in the individual while they were already IBD in at least one ancestor, whereas $F_{NEW}$ is the probability that alleles are IBD for the first time in the individual's pedigree [8].

To calculate $F_{ANC}$ and $F_{NEW}$ (and other inbreeding coefficients), a gene dropping based algorithm has been developed and implemented in GRain software [10]. The GRain algorithm has also been incorporated in the PEDIG package [11], in versions 2007 and later. Various studies have used the GRain algorithm, either in GRain itself [12–16] or in PEDIG [17–19], to calculate $F_{ANC}$ and $F_{NEW}$.

The objective of this study was to demonstrate that the previous version of GRain (v 2.1) produced biased estimates of $F_{ANC}$ and $F_{NEW}$. For several simple pedigrees, we show how $F_{ANC}$ and $F_{NEW}$ can be calculated by hand. We also investigate the magnitude of the bias for two example data sets of Hungarian Pannon White rabbits and Dutch Holstein Friesian dairy cattle. A revised version of GRain software (v 2.2), which provides unbiased $F_{ANC}$ and $F_{NEW}$ estimates, is now available online.

## 2. Calculation of Ancestral and New Inbreeding Coefficients by Hand

For simple pedigrees, Kalinowski's ancestral inbreeding ($F_{ANC,X}$) and new inbreeding ($F_{NEW,X}$) coefficients of an individual X can be calculated by hand. To do so, Mendelian inheritance principles are followed, meaning that each allele has a probability of 0.5 to be passed on from parent to offspring. First, Wright's classical inbreeding coefficient ($F_X$) is determined. The $F_X$ is defined as the probability that the two alleles at a random locus in individual X are IBD, and is calculated as [1]:

$$F_X = \sum_{i=1}^{n}(1 + F_i)\left(\frac{1}{2}\right)^{k_s+k_d+1} \tag{1}$$

where $n$ is the number of paths connecting the sire of X with the dam of X through the $i$th common ancestor, $F_i$ is the inbreeding coefficient of the $i$th common ancestor, and $k_s$ and $k_d$ are the number of generations from, respectively, sire and dam (included) to the $i$th common ancestor (excluded). Then, $F_{ANC,X}$ is calculated as the probability that X is IBD for an allele, given that this allele was also IBD in at least one of the ancestors of X. Finally, $F_{NEW,X}$ is obtained by subtracting $F_{ANC,X}$ from $F_X$, since the ancestral and new inbreeding sum up to the total inbreeding.

In Figure 1, four example pedigrees are shown. The corresponding inbreeding coefficients are provided in Table 1. In example (1), the $F_X$ equals 0.0078 ($0.5^7$), because there is a single path that connects parents F and G through common ancestor A, which is of length 7 ($k_s + k_d + 1 = 7$), and ancestor A is non-inbred ($F_A = 0$). The $F_{ANC,X}$ for this example is 0, because none of the ancestors of X are inbred. Consequently, $F_{NEW,X}$ is equal to $F_X$ (so 0.0078).

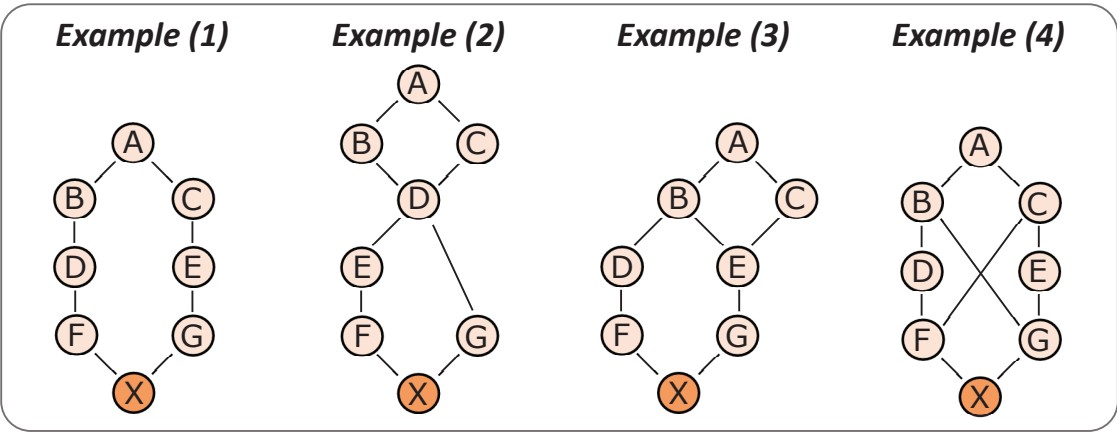

**Figure 1.** Example pedigrees for the calculation of classical and Kalinowski's inbreeding coefficients. X is the individual of interest and the other letters represent ancestors of X. Inbreeding coefficients for individual X, corresponding to the four pedigrees, are shown in Table 1.

**Table 1.** Inbreeding coefficients for four example pedigrees (Figure 1), estimated with revised and previous version of GRain.

| Pedigree | $F_X$ | Revised Version (v 2.2) | | Previous Version (v 2.1) | | Difference in $F_{ANC,X}$ |
|---|---|---|---|---|---|---|
| | | $F_{ANC,X}$ | $F_{NEW,X}$ | $F_{ANC,X}$ | $F_{NEW,X}$ | |
| (1) | 0.0078 | 0 | 0.0078 | 0 | 0.0078 | 0 |
| (2) | 0.0703 | 0.0156 | 0.0547 | 0.0156 | 0.0547 | 0 |
| (3) | 0.0390 | 0.0078 | 0.0312 | 0.0039 | 0.0351 | 0.0039 |
| (4) | 0.1641 | 0.0390 | 0.1250 | 0.0234 | 0.1406 | 0.0156 |

$F_X$, classical inbreeding coefficient of individual X; $F_{ANC,X}$, Kalinowski's ancestral inbreeding coefficient of individual X; $F_{NEW,X}$, Kalinowski's new inbreeding coefficient of individual X.

In example (2), the $F_X$ equals 0.0703, because it is the inbreeding on ancestor D ($0.5^4$) multiplied with $(1 + F_D)$, where $F_D$ is the inbreeding coefficient of ancestor D ($0.5^3$). The $F_{ANC,X}$ is calculated as the probability that X is IBD for an allele that was IBD in D as well. Since D is the only inbred ancestor, we do not need to consider the IBD status of any other ancestors. The probability that D is IBD for an allele from its grandparent A, is the inbreeding coefficient of D on A and equals 0.125 ($0.5^3$). To obtain $F_{ANC,X}$, this probability has to be multiplied with the probabilities that the allele is passed on to X, through both the paths D-E-F-X and D-G-X. The probability that E inherits the allele from D is simply 1, because D is IBD. The probability that F inherits the allele from E is 0.5 and that X inherits it from F is also 0.5, so the total probability for the path D-E-F-X is 0.25 ($0.5^2$). Similarly, the probability for path D-G-X is 0.5. This gives a total probability of $0.125 \times 0.25 \times 0.5 = 0.0156$ for $F_{ANC,X}$. Consequently, $F_{NEW,X} = F_X - F_{ANC,X} = 0.0703 - 0.0156 = 0.0547$. Note that, in this example, the $F_{ANC,X}$ can also be calculated as two times the inbreeding coefficient of X on D ($0.5^4$), multiplied with the inbreeding coefficient of D on A ($0.5^3$). However, it is important to realize that this reasoning only holds for scenarios in which one inbreeding loop is "on top of the other", and not when there is an overlap in inbreeding loops, such as in examples (3) and (4).

In example (3), the $F_X$ equals 0.0390 and is the sum of inbreeding on ancestor A ($0.5^7$) and on ancestor B ($0.5^5$). The $F_{ANC,X}$ is calculated as the probability that X is IBD for an allele that was IBD in ancestor E as well. Since ancestor E is the only inbred ancestor, we do not need to consider the IBD status of any other ancestors. The probability that E is IBD for an allele from its grandparent A, is the inbreeding coefficient of E on A and equals 0.125 ($0.5^3$). This probability has to be multiplied by the probability that this allele is passed on to X through both the path E-G-X and B-D-F-X. The probability that G inherits the allele from E is 1, because E is IBD. The probability that X inherits the allele from G is 0.5, so the total probability for the path E-G-X is 0.5. The probability that B carries the allele is 1,

otherwise E could not have been IBD. The probability that the allele is passed on from B to D to F and to X is 0.125 ($0.5^3$). This gives a total probability of $0.125 \times 0.125 \times 0.5 = 0.0078$ for $F_{ANC,X}$. Consequently, $F_{NEW,X} = F_X - F_{ANC,X} = 0.0390 - 0.0078 = 0.0312$.

In example (4), the $F_X$ equals 0.1641 and is the sum of inbreeding on ancestor A ($0.5^7 + 0.5^5$), on ancestor B ($0.5^4$) and on ancestor C ($0.5^4$). The $F_{ANC,X}$ in this example is the probability that X is IBD for an allele that was also IBD in F and/or G (since F and G are inbred ancestors). The $F_{ANC,X}$ is the sum of the probabilities for three scenarios: (i) X is IBD for an allele that was IBD in both F and G, (ii) X is IBD for an allele that was IBD in F, but not in G, and (iii) X is IBD for an allele that was IBD in G, but not in F. The probability that F is IBD for an allele from A is the inbreeding coefficient of F on A and equals 0.0625 ($0.5^4$). If F is IBD for an allele from A, then both B and C must be carriers of that allele, and the probability that G is also IBD for that same allele is 0.125 ($0.5^3$), since this is the probability that G inherits that allele through B-G (0.5) multiplied with the probability that G inherits that allele through C-E-G ($0.5^2$). When F and G are IBD for the same allele, X has to be IBD for that allele as well. Therefore, the probability that scenario (i) happens is 0.0078 (i.e., $0.0625 \times 0.125 \times 1$). If F is IBD for an allele from A, the probability that G carries two other "unknown" alleles is 0.375 (i.e., $0.5 \times (1 - 0.5^2)$), leaving $1 - 0.125 - 0.375 = 0.5$, for the probability that G carries one copy of the allele and one copy of an unknown allele (scenario ii). In that case, the probability that the allele is inherited by X from G is 0.5. The total probability for scenario (ii) is therefore 0.0156 (i.e., $0.0625 \times 0.5 \times 0.5$). Due to the symmetry in the pedigree, the probability for scenario (iii) is equal to that of scenario (ii), so 0.0156. Thus, the total probability that X is IBD for an allele that was also IBD in F and/or G, i.e., the $F_{ANC,X}$, equals $0.0078 + 0.0156 + 0.0156 = 0.0391$. Consequently, $F_{NEW,X} = F_X - F_{ANC,X} = 0.1641 - 0.0391 = 0.1250$.

## 3. Underestimation of Ancestral Inbreeding by Previous Version of GRain

In GRain, a stochastic approach known as gene dropping [20] is implemented to calculate inbreeding coefficients. In this approach, many independent simulations are run. In each simulation, alleles are dropped through the pedigree following Mendelian inheritance rules, and the IBD-status of individuals is stored. After all simulations are completed, the $F_X$ is estimated as the fraction of simulations in which the alleles of individual X were IBD. Similarly, the $F_{ANC,X}$ is calculated as the fraction of simulations in which X was IBD for an allele that was already IBD in one of the ancestors of X. The accuracy of the estimated inbreeding coefficients is higher when more simulations are run. As shown by Baumung et al. [10], using $10^6$ simulations provides estimates of inbreeding coefficients that show a correlation of >0.999 with inbreeding coefficients calculated using a deterministic approach (with only minor differences at the fourth decimal). A more detailed explanation of the GRain program and its computational demands is given by Baumung et al. [10].

When $F_{ANC,X}$ was computed using the previous version of GRain (v 2.1), the $F_{ANC,X}$ for examples (1), (2), (3) and (4) from Figure 1 equaled 0, 0.0156, 0.0039 and 0.0234, respectively (Table 1). Although the coefficients for examples (1) and (2) were correct, the $F_{ANC,X}$ coefficients for examples (3) and (4) were underestimated. Note that example (3) is equivalent to the example used by McParland et al. [18], in Figure 1 in their paper, for which they reported the incorrect $F_{ANC,X}$ estimate of 0.0039.

The underestimation of $F_{ANC,X}$ was occasionally caused by an incorrect tracking of IBD-status of ancestors throughout the pedigree. In the previous version of GRain (v 2.1), every individual was given a flag that indicated whether one of their ancestors had been IBD (1 if true, 0 if false). This flag was calculated as the sum of the flags of the parents, divided by two. Thus, when both parents had a flag of 1, the flag of the offspring would also be 1, which is correct. However, when only one of the parents had a flag of 1 (and the other 0), the offspring would get a value of 0.5, which is incorrect (since it should be 1). In the revised version of GRain (v 2.2), this issue was solved by obtaining the flag of an offspring as the maximum of the flags of its parents.

To clarify, in example (2) in Figure 1, whenever ancestor D was IBD, both parents F and G had a flag of 1 and X also got a flag of 1. Therefore, the $F_{ANC,X}$ was estimated correctly. In example (3), however, whenever ancestor E was IBD, parent G had a flag of 1 and parent F had a flag of 0 and, as a

result, X got a flag of 0.5. Consequently, for simulations in which individual X was IBD for an allele that was also IBD in E, a value of 0.5 was stored (instead of 1) for the $F_{ANC,X}$ calculation. After simulations were completed, the stored values were summed across simulations and divided by the total number of simulations. Since stored values were underestimated by a factor two, the $F_{ANC,X}$ for example (3) was also underestimated by a factor two. In example (4), whenever both F and G were IBD, X got a flag of 1. This happened in 0.0078 of the simulations (see explanation in the previous section for calculation by hand, scenario (i)). When only parent F or parent G were IBD, while the other parent was not, X got a flag of 0.5. This happened in 0.0156 + 0.0156 = 0.0312 of the simulations (see explanation in the previous section for calculation by hand, scenarios (ii) and (iii)). Therefore, the $F_{ANC,X}$ for example (4) was underestimated by some factor between one and two. More specifically, the underestimated $F_{ANC,X}$ was equal to 0.0078 + (0.5 × 0.0312) = 0.0234.

## 4. Examples for Pannon White Rabbits and Holstein Friesian Cattle

To investigate the impact of the incorrect estimation, we computed $F_{ANC}$ and $F_{NEW}$ for two example data sets, using both the previous and revised version of GRain, and $10^6$ simulations. The first data set was a pedigree of 22,781 rabbits of the Hungarian Pannon White (PW) breed. This pedigree included 6760 rabbits (1421 bucks and 5339 does) with offspring and 16,021 rabbits without offspring. All rabbits were born between 1992 and 2016. To assess pedigree completeness, the number of complete generations (NCG) and the complete generation equivalent (CGE) were computed for each rabbit. The CGE was computed as the sum of $(1/2)^n$ of all known ancestors of an individual, with $n$ being the number of generations between the individual and a given ancestor. The mean NCG in the PW pedigree was 4.0 (ranging from 0 to 10) and the mean CGE was 8.6 (ranging from 0 to 22.1). The second data set contained 37,061 Dutch Holstein Friesian (HF) cows, which were part of a larger pedigree of 167,924 individuals (19,363 bulls and 148,561 cows) and were used by Doekes et al. [21]. These HF cows were born between 2012 and 2016 and were filtered to have a high pedigree completeness (NCG ≥ 3 and CGE ≥ 10), and have phenotypic information on 305-day milk, fat and protein yields. The mean NCG in these HF cows was 6.5 generations (ranging from 3 to 9) and the mean CGE was 12.5 generation equivalents (ranging from 10.0 to 14.7). More details on the HF data set can be found in Doekes et al. [21].

For both the PW and HF data set, the total inbreeding coefficients (*F*) were identical across the previous and revised version of GRain. The $F_{ANC}$ in the previous version however, was generally underestimated and the $F_{NEW}$ was overestimated (Figure 2). For the PW data set and for inbreeding coefficients above zero, the $F_{ANC}$ from the previous version was on average 0.65 times the revised $F_{ANC}$ (and the $F_{NEW}$ was 1.27 times the revised $F_{NEW}$). For the HF data set and inbreeding coefficients above zero, the $F_{ANC}$ from the previous version was on average 0.71 times the revised $F_{ANC}$ (and the $F_{NEW}$ was 1.36 times the revised $F_{NEW}$). Pearson correlation coefficients between coefficients estimated with the previous and revised version were high. For the PW data set, the correlations between the previous and revised version equaled 0.997 and 0.968 for $F_{ANC}$ and $F_{NEW}$, respectively. For the HF data set, these correlations equaled 0.993 and 0.987, respectively. This indicates that the underestimation of $F_{ANC}$ (and overestimation of $F_{NEW}$) did not strongly affect the ranking of animals.

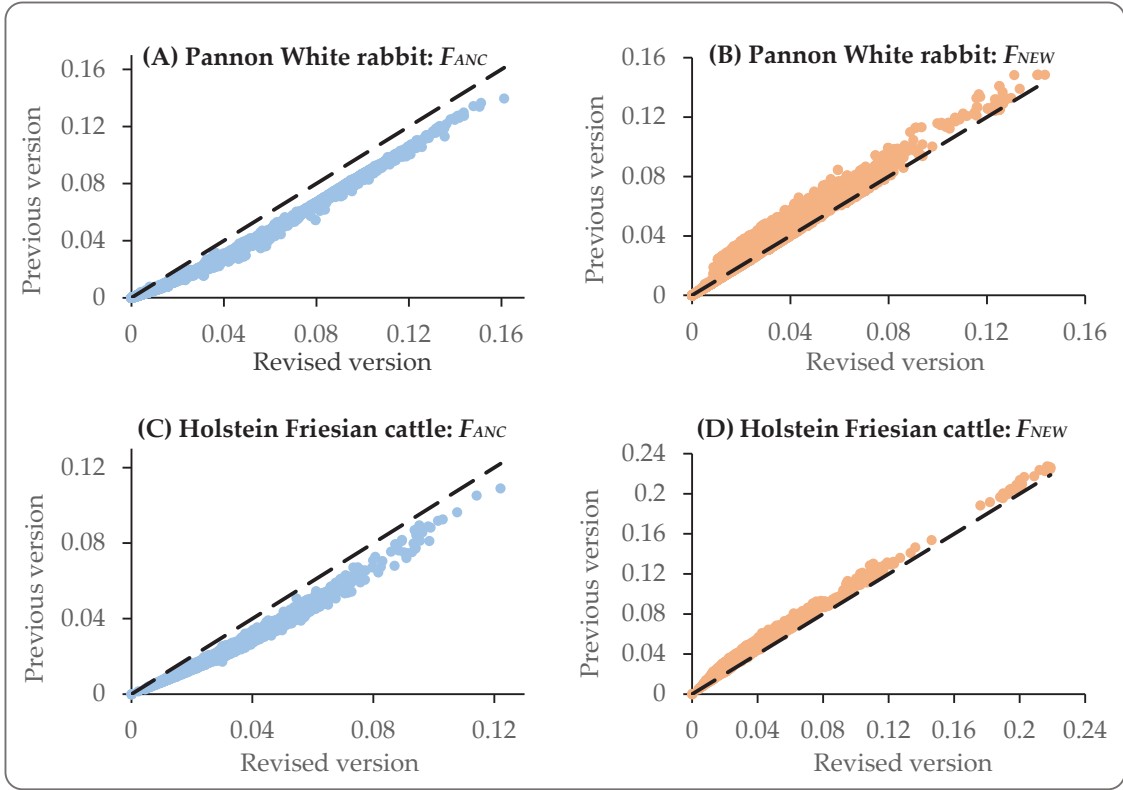

**Figure 2.** Relationship between Kalinowski's inbreeding coefficients calculated with previous (v 2.1) and revised (v 2.2) version of GRain, for example data sets of Pannon White rabbits (PW; $n$ = 22,781) and Holstein Friesian cattle (HF; $n$ = 37,061). The dashed line indicates $y = x$, i.e., a relationship in which there is no difference in estimation between the two GRain versions. $F_{ANC}$: Kalinowski's ancestral inbreeding coefficient. $F_{NEW}$: Kalinowski's new inbreeding coefficient. (**A**) $F_{ANC}$ for the PW data set, (**B**) $F_{NEW}$ for the PW data set, (**C**) $F_{ANC}$ for the HF data set, and (**D**) $F_{NEW}$ for the HF data set.

For the Holstein Friesian data set, we also investigated the potential differences in inbreeding depression estimates for $F_{ANC}$ and $F_{NEW}$, calculated with the previous and revised version of GRain. A linear mixed model was run in ASReml 4.1 [22], in which $F_{ANC}$ and $F_{NEW}$ were fitted as fixed effects and the regression coefficients on $F_{ANC}$ and $F_{NEW}$ were used as estimates of inbreeding depression (see Doekes et al. [21] for a detailed explanation). In general, differences between inbreeding depression estimates based on the previous and revised version of GRain were small (Figure 3). For example, the effect of a 1% increase in $F_{NEW}$ on 305-day milk yield was −46.4 kg (SE = 12.4 kg) for the previous version and −47.3 kg (SE = 11.2 kg) for the revised version. Standard errors for the inbreeding depression effects appeared smaller when the revised version was used to estimate $F_{ANC}$ and $F_{NEW}$, compared to when the previous version was used. For example, the mean standard error of inbreeding depression estimates for fat and protein yields was 0.51 kg for the revised version, and 0.67 kg for the previous version. The overall conclusion, that $F_{NEW}$ was associated with significant inbreeding depression, while $F_{ANC}$ was not, was the same for both versions. Based on these findings, we expect that conclusions from other studies using $F_{ANC}$ and $F_{NEW}$ estimates from GRain v 2.1 (e.g., [17,18]) will also largely hold. However, they should be interpreted with caution.

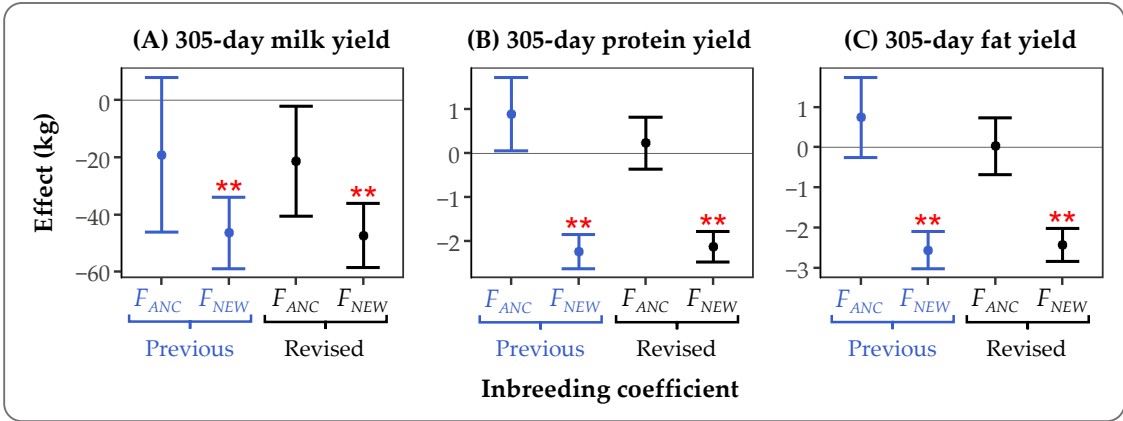

**Figure 3.** Effect of a 1% increase in Kalinowski's ancestral ($F_{ANC}$) and new ($F_{NEW}$) inbreeding on yield traits in Dutch Holstein Friesian cattle ($n$ = 37,061), for $F_{ANC}$ and $F_{NEW}$ calculated with the previous (v 2.1) and revised (v 2.2) version of GRain. Red asterisks indicate effects that significantly ($p < 0.001$) differed from zero. (**A**) 305-day milk yield, (**B**) 305-day protein yield, and (**C**) 305-day fat yield.

## 5. Conclusions

The previous version of GRain software (v 2.1) systematically underestimated Kalinowski's ancestral inbreeding and, consequently, overestimated Kalinowski's new inbreeding coefficients. Although the magnitude of bias was rather small, results from studies based on biased estimates should be interpreted with caution. The GRain software has been revised, and the revised version (v 2.2), which provides unbiased estimates of Kalinowski's coefficients, can be downloaded from [23] or [24].

**Author Contributions:** H.P.D. and J.J.W. performed the Holstein Friesian data analysis; J.F. and G.K. performed the Pannon White data analysis; H.P.D. prepared the manuscript. H.P.D., I.C., I.N., J.F., G.K. and J.J.W. participated in the interpretation of results and revision of the manuscript. All authors have read and agreed to the published version of the manuscript.

**Funding:** Calculations performed on the Dutch Holstein Friesian population were conducted as part of the IMAGE project, which received funding from the European Union's Horizon 2020 Research and Innovation Programme under the grant agreement no 677353. Calculations performed on the Pannon White rabbit population were supported by the Hungarian Scientific Research Fund (OTKA) K 128177 project. The study was co-funded by the Dutch Ministry of Agriculture, Nature and Food Quality (KB-34-013-002).

**Acknowledgments:** The authors would like to thank the Dutch-Flemish cattle improvement cooperative (CRV) for providing the Holstein Friesian data.

**Conflicts of Interest:** The authors declare no conflict of interest.

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
