# Peer review of "Revised Calculation of Kalinowski’s Ancestral and New Inbreeding Coefficients"

_diversity, doi:10.3390/d12040155_

Round 1

Reviewer 1 Report

The manuscript is detailed and well written. It is really interesting for people that are interested in the details of inbreeding coefficient calculations, and also for people that are interested in partitioning the inbreeding in ancient inbreeding and recent inbreeding. I have particularly appreciated the effort provided by Authors to explain in a simple way the inbreeding coefficient calculation, to provide the basis to understand the theory about ancient inbreeding and recent inbreeding. Aiming to have a technical note as clearer as possible, I have reported some comments that are basically done to increase the comprehension in the reader, both in the first theoretical part and in the following practical example. This last part, in particular, lacks in details that in my opinion are important to report since this is a technical note. However, I think that the manuscript could be then accepted after a minor revision of these issues.

L67-70: I recommend to revise the formula of Wright’s Inbreeding coefficient using the original notation of Sewall Wright (1922), that is by writing “k+k’+1” instead of “k+1”, and specifying that k and k’ are the number of generations respectively from sire and dam (included) to the ancestor in question (excluded). I recommend this because if you talk about steps in the path, then the proper formula includes “k-1” and not “k+1”: e.g., in Example 1, you have 8 steps: from X to F, F to D, D to B, B to A, A to C, C to E, E to G, G to X. Therefore you should do k-1. In some texts you can find this kind of notation. In some other texts you can find just “k”, where k is the number of ancestors of X. Therefore, I suggest to change your notation to the original one of Wright to avoid misunderstandings in people that are not familiar with inbreeding path coefficients calculation (or to use the notation of “k-1” if you want to talk about paths).

L91: The explanation is clear, compliments for this. I just suggest to add that the probability for each path is 0.5^a, where a is the number of ancestors in the path excluding X and the target inbred ancestor (D). Obviously you can use different letters than a and a different expression, but I think that this specification could improve the clearness of this part.

L97: You could also specify at the end of the last sentence that this value is also the inbreeding coefficient for E (the same for L109 and the inbreeding coefficient for F).

L99: I think it is more correct to indicate the second path as E-B-D-F-X. But if you want to include just descendants and not ascendants in the path, B-D-F-X is ok.

L110-111: How did you compute the 0.125? I saw it is 0.5^3, is this because you moved from A (probability = 1 to have the same allele of F) to C (probability of 0.5), to E (probability of 0.5), to G (probability of 0.5), therefore 0.5^3. Is this correct? If yes, could you add in brief this information.

L137-138: Is the underestimation due to the fact that you multiply the resulting 0.0078 of FANC,X by 0.5 due to the flag? Could you specify this here (that is, what is the point in the calculation in which the flags are used by the previous version of GRain).

L141-142: How do the simulations work? You have provided detailed explanations of the calculation by hand, but how is the program working with complex pedigree? Is the pedigree of Example 4 already too complex for the calculation by hand of the program? Could you write a couple of lines about simulations?

L147: This point is similar to the previous issue: why did you ask 10^6 replications to the programs? Do you need to average the result of the replications for obtaining the final estimate? How does the inbreeding estimation work? I understand that these requests are referred to the technical features of the program, but since this is a technical note about this program, I think it is important to provide some detailed information about it.

L163-164: How did you assessed the inbreeding depression? Could you briefly report some methodological details?

L164-166 and Figure 3: Did you find significant differences among the two versions? Or the significance is just related to the effect of FANC and FNEW within program’s version? Could you report in the caption of Figure 3 that asterisks indicate significant differences between the inbreeding depression effects of FANC and FNEW?

L166-168: Could you provide in brackets some indicative values for the standard errors obtained using the two versions (e.g., average values for each version)?

L168-169: Do you think that with some kinds of pedigrees the two versions of the software could provide different conclusions? Moreover: at the beginning of the manuscript you have mentioned PEDIG software; are the conclusions provided by this software also biased? Have you tried to find out FANC and FNEW also using this software? It could be really interesting to say if FANC and FNEW provided by the two software are similar (and, in case, to see if PEDIG estimates are similar to v2.1 or to v2.2 of GRain). Could you provide some evidences about this issue?

L180-181: Do you think that all the studies that have used v2.1 (and I guess also earlier versions) of the GRain software have provided incorrect conclusions, or the overall conclusions about FANC and FNEW are the same for both version, as for your datasets? Did you try to test the program also with other datasets?

As I reported in the comments to Authors, the manuscript is well written, just requiring some methodological explanations. Some additional results could be also interesting to see (comparison of the Author findings using GRain software and eventual finding using PEDIG software). I think the manuscript could be suitable for publication after minor revision.

Author Response

Author response reviewer 1

The manuscript is detailed and well written. It is really interesting for people that are interested in the details of inbreeding coefficient calculations, and also for people that are interested in partitioning the inbreeding in ancient inbreeding and recent inbreeding. I have particularly appreciated the effort provided by Authors to explain in a simple way the inbreeding coefficient calculation, to provide the basis to understand the theory about ancient inbreeding and recent inbreeding. Aiming to have a technical note as clearer as possible, I have reported some comments that are basically done to increase the comprehension in the reader, both in the first theoretical part and in the following practical example. This last part, in particular, lacks in details that in my opinion are important to report since this is a technical note. However, I think that the manuscript could be then accepted after a minor revision of these issues.

>AU: Thank you for your kind words and the useful comments that have helped to improve the manuscript. Based on your comments, we have made various changes in the manuscript (highlighted in yellow). Please find a point-by-point answer to your comments below.  

L67-70: I recommend to revise the formula of Wright’s Inbreeding coefficient using the original notation of Sewall Wright (1922), that is by writing “k+k’+1” instead of “k+1”, and specifying that k and k’ are the number of generations respectively from sire and dam (included) to the ancestor in question (excluded). I recommend this because if you talk about steps in the path, then the proper formula includes “k-1” and not “k+1”: e.g., in Example 1, you have 8 steps: from X to F, F to D, D to B, B to A, A to C, C to E, E to G, G to X. Therefore you should do k-1. In some texts you can find this kind of notation. In some other texts you can find just “k”, where k is the number of ancestors of X. Therefore, I suggest to change your notation to the original one of Wright to avoid misunderstandings in people that are not familiar with inbreeding path coefficients calculation (or to use the notation of “k-1” if you want to talk about paths).

>AU: We now use “ks+ kd+1” and specify what ks+ kd are (L68-71). We believe that ks and kd are more informative than k and k’ (especially for someone who is unfamiliar with the method). For the first example, we now also explain the use of the path counting method in more detail (L76-78).

L91: The explanation is clear, compliments for this. I just suggest to add that the probability for each path is 0.5^a, where a is the number of ancestors in the path excluding X and the target inbred ancestor (D). Obviously you can use different letters than a and a different expression, but I think that this specification could improve the clearness of this part.

>AU: We have added this explanation, although this reasoning only works for example 2 (L96-100).

L97: You could also specify at the end of the last sentence that this value is also the inbreeding coefficient for E (the same for L109 and the inbreeding coefficient for F).

 >AU: We have specified this in three instances (L90, L104-105, and L117-118). 

L99: I think it is more correct to indicate the second path as E-B-D-F-X. But if you want to include just descendants and not ascendants in the path, B-D-F-X is ok.

>AU: We have decided to stick with B-D-F-X. We recognize that for the path counting context the use of E-B-D-F-X might be more attractive. However, this would be problematic in the explanation, because there is no inheritance from E to B. Also, since E is IBD, we already know that B carries the allele (as explained in L108-109).   

L110-111: How did you compute the 0.125? I saw it is 0.5^3, is this because you moved from A (probability = 1 to have the same allele of F) to C (probability of 0.5), to E (probability of 0.5), to G (probability of 0.5), therefore 0.5^3. Is this correct? If yes, could you add in brief this information.

>AU: We have added an explanation on why it is 0.53 (L118-121). For the calculation, we apply the reasoning that if F is IBD for an allele, let’s say allele_1, that B and C must be carriers of allele_1. Then, we can calculate the probability that G becomes IBD for allele_1 by multiplying the probabilities that G inherits allele_1 from B (0.5) with the probability that G inherits allele_1 through C-E-G (0.52), which results in 0.5*0.52 = 0.53.  

L137-138: Is the underestimation due to the fact that you multiply the resulting 0.0078 of FANC,X by 0.5 due to the flag? Could you specify this here (that is, what is the point in the calculation in which the flags are used by the previous version of GRain).

>AU: We added an explanation (L158-162) which, in combination with the new paragraph on how the simulations work (L131-141), hopefully is sufficient.

L141-142: How do the simulations work? You have provided detailed explanations of the calculation by hand, but how is the program working with complex pedigree? Is the pedigree of Example 4 already too complex for the calculation by hand of the program? Could you write a couple of lines about simulations?

>AU: We have added a paragraph on how the simulations work (L131-141).

L147: This point is similar to the previous issue: why did you ask 10^6 replications to the programs? Do you need to average the result of the replications for obtaining the final estimate? How does the inbreeding estimation work? I understand that these requests are referred to the technical features of the program, but since this is a technical note about this program, I think it is important to provide some detailed information about it.

>AU: See paragraph on simulations (L131-141). 

L163-164: How did you assessed the inbreeding depression? Could you briefly report some methodological details?

>AU: We have added some methodology (L202-205) and refer to previous work for a more detailed explanation.

L164-166 and Figure 3: Did you find significant differences among the two versions? Or the significance is just related to the effect of FANC and FNEW within program’s version? Could you report in the caption of Figure 3 that asterisks indicate significant differences between the inbreeding depression effects of FANC and FNEW?

>AU: Thank you for this comment. We forgot to mention that asterisks indicate significant inbreeding depression (i.e. significantly different from zero), which is now included in the figure caption (L218-219). Significance testing between the revised and previous version would be tricky, because the estimates are obtained from 2 independent linear mixed models (see e.g. ASReml manual).

 L166-168: Could you provide in brackets some indicative values for the standard errors obtained using the two versions (e.g., average values for each version)?

 >AU: We have included SEs in the text (L207-212).

L168-169: Do you think that with some kinds of pedigrees the two versions of the software could provide different conclusions? Moreover: at the beginning of the manuscript you have mentioned PEDIG software; are the conclusions provided by this software also biased? Have you tried to find out FANC and FNEW also using this software? It could be really interesting to say if FANC and FNEW provided by the two software are similar (and, in case, to see if PEDIG estimates are similar to v2.1 or to v2.2 of GRain). Could you provide some evidences about this issue?

>AU: As shown for the 4 simple example pedigrees, there may be differences in the degree of underestimation across individuals. This will depend on whether inbred ancestors occur only on one side (e.g. only the sire side) or on both sides (sire and dam side) of the individual’s pedigree. For most populations, we expect that there will be some animals with inbred ancestors on both sides and some with inbred ancestors on one side. Exceptions may be crossbreeding programs (e.g. pig and poultry). For these pedigrees the FX of the crossbred is expected to be zero and, thus, also the FANC,X and FNEW,X  would be zero. We decided not to speculate on these kind of differences in this technical note.

>AU: Part of the exact code of GRain v2.1 (among others the part for calculating FANC and FNEW) is actually incorporated in PEDIG, which can be downloaded from https://www6.jouy.inrae.fr/gabi_eng/Our-resources/Tool-development/Pedig . We have now specified this in the manuscript (L54-56). We have also notified the author of PEDIG. 

L180-181: Do you think that all the studies that have used v2.1 (and I guess also earlier versions) of the GRain software have provided incorrect conclusions, or the overall conclusions about FANC and FNEW are the same for both version, as for your datasets? Did you try to test the program also with other datasets?

>AU: We expect that, based on our findings, these conclusions will be largely the same with v2.1 and 2.2. We now explicitly state this expectation in the manuscript (L213-215). In an additional inbreeding depression study for insect bite hypersensitivity in Kladrub horses, significance did not change after correcting calculations of FNEW. We did not check other data sets from previous studies (which we refer to in the manuscript), because these were not publicly available. We believe that we covered quite a wide range of inbreeding coefficients with the data sets that we did use.

As I reported in the comments to Authors, the manuscript is well written, just requiring some methodological explanations. Some additional results could be also interesting to see (comparison of the Author findings using GRain software and eventual finding using PEDIG software). I think the manuscript could be suitable for publication after minor revision.

Reviewer 2 Report

This manuscript is very interesting. I recommend publish this manuscript in present form. This manuscript is very interesting. Explaining the fixes in a computer program is very beneficial. In particular, justification of the underestimation of the ancestral inbreeding coefficient. I very positively evaluate the demonstration of the methodological procedure in estimating Kalinowski ancestral inbreeding on individual exemplary pedigrees. In addition, examples are supplemented by estimates on real populations. I recommend publish this manuscript in present form.

Author Response

Author response reviewer 2

This manuscript is very interesting. I recommend publish this manuscript in present form. This manuscript is very interesting. Explaining the fixes in a computer program is very beneficial. In particular, justification of the underestimation of the ancestral inbreeding coefficient. I very positively evaluate the demonstration of the methodological procedure in estimating Kalinowski ancestral inbreeding on individual exemplary pedigrees. In addition, examples are supplemented by estimates on real populations. I recommend publish this manuscript in present form.

>AU: Thank you for your kind comments. Based on the comments of the other reviewers, we have made various changes to the manuscript (which are highlighted in yellow).

Reviewer 3 Report

Dear authors,

this is an interesting technical note. However, some minor changes should be done and the part dealing with real data must be improved. The most critical point here is, that you only give the total number of animals in the pedigree.

Abstract:

Please give more information about the real data. At least number of animals in the pedigree.

Introduction

Line 30 – 38: It is not clear that you are mainly talking about the classical concept of Wright. This should be clarified.

Line 53 – 51: Please give some more information about the original papers of Ballou and Kallinowski, eg what are the animals these authors are working with.

Line 145 ff.: Please give more information about the real pedigree data, e.g number of males and females, how many generations of pedigree . What is the pedigree completness. I also can not understand how you extract animals in the Holstein pedigree. All these facts make the interpretation of the results difficult.

This is a good tecnical note. However, I would like to see more information about the real data. The correlations betweeen the estimates of the old and new version of the software are high. However, the concept of ancestral inbreeding should be more applied and therefore we need more studies that people get to know about this.

Author Response

Author response reviewer 3

Dear authors, this is an interesting technical note. However, some minor changes should be done and the part dealing with real data must be improved. The most critical point here is, that you only give the total number of animals in the pedigree.

>AU: Thank you for your kind words and the useful comments that have helped to improve the manuscript. Based on your comments, we have made various changes in the manuscript (highlighted in yellow). Please find a point-by-point answer to your comments below.  

Abstract:

Please give more information about the real data. At least number of animals in the pedigree.

>AU: We have added the number of individuals to the abstract (L24) and added more information about the pedigrees in the body of the text (see below).

Introduction

Line 30 – 38: It is not clear that you are mainly talking about the classical concept of Wright. This should be clarified.

>AU: We now indicate that we talk about pedigree inbreeding as introduced by Wright (L32-33)

Line 53 – 51: Please give some more information about the original papers of Ballou and Kallinowski, eg what are the animals these authors are working with.

>AU: We now indicate what Ballou and Kallinowski were working on (L44 and L47-48).

Line 145 ff.: Please give more information about the real pedigree data, e.g number of males and females, how many generations of pedigree . What is the pedigree completness. I also can not understand how you extract animals in the Holstein pedigree. All these facts make the interpretation of the results difficult.

>AU: We now report the numbers of males and females in the pedigree, report commonly used measures of pedigree completeness and report why these Holstein Friesian cows were used (L172-185).

This is a good tecnical note. However, I would like to see more information about the real data. The correlations betweeen the estimates of the old and new version of the software are high. However, the concept of ancestral inbreeding should be more applied and therefore we need more studies that people get to know about this.